# A Novel Approach to Optimize the Fabrication Conditions of Thin Film Composite RO Membranes Using Multi-Objective Genetic Algorithm II

**DOI:** 10.3390/polym12020494

**Published:** 2020-02-24

**Authors:** Fekri Abdulraqeb Ahmed Ali, Javed Alam, Arun Kumar Shukla, Mansour Alhoshan, Basem M. A. Abdo, Waheed A. Al-Masry

**Affiliations:** 1Chemical Engineering Department, College of Engineering, King Saud University, P.O. Box-2455, Riyadh 11451, Saudi Arabia; falhulidy@ksu.edu.sa (F.A.A.A.); mhoshan@ksu.edu.sa (M.A.); walmasry@ksu.edu.sa (W.A.A.-M.); 2King Abdullah Institute for Nanotechnology, King Saud University, P.O. Box- 2455, Riyadh 11451, Saudi Arabia; ashukla@ksu.edu.sa; 3K.A. CARE Energy Research and Innovation Center at Riyadh, Riyadh 11451, Saudi Arabia; 4Advanced Manufacturing Institute, King Saud University, P.O. Box- 2455, Riyadh 11451, Saudi Arabia

**Keywords:** thin film composite membrane, optimize the fabrication conditions, multi-objective genetic algorithm II, response surface model, salt rejection

## Abstract

This work focuses on developing a novel method to optimize the fabrication conditions of polyamide (PA) thin film composite (TFC) membranes using the multi-objective genetic algorithm II (MOGA-II) method. We used different fabrication conditions for formation of polyamide layer—trimesoyl chloride (TMC) concentration, reaction time (t), and curing temperature (Tc)—at different levels, and designed the experiment using the factorial design method. Three functions (polynomial, neural network, and radial basis) were used to generate the response surface model (RSM). The results showed that the radial basis predicted good results (R^2^ = 1) and was selected to generate the RSM that was used as the solver for MOGA-II. The experimental results indicate that TMC concentration and t have the highest influence on water flux, while NaCl rejection is mainly affected by the TMC concentration, t, and Tc. Moreover, the TMC concentration controls the density of the PA, whereas t confers the PA layer thickness. In the optimization run, MOGA-II was used to determine optimal parametric conditions for maximizing water flux and NaCl rejection with constraints on the maximum acceptable levels of Na_2_SO_4_, MgSO_4_, and MgCl_2_ rejections. The optimized solutions were obtained for longer t, higher Tc, and different TMC concentration levels.

## 1. Introduction

Most current nanofiltration (NF) and reverse osmosis (RO) membranes are prepared as thin film composite (TFC) structures composed of three distinct layers: A porous, non-woven, polyester support layer; an intermediate layer made of polysulfone (PSF); and a thin polyamide top layer that is 100–200 nm thick [1]. The polyamide (PA) layer is synthesized by an interfacial polymerization (IP) reaction between a diamine (e.g., m-phenylenediamine, MPD) and acid chloride monomers (e.g., trimesoyl chloride and TMC) at the surface of a PSF intermediate layer [2,3,4]. In the IP process, the immiscible aqueous MPD solution and the organic TMC solution are combined, and a thin PA film immediately forms between the two phases on the organic TMC side of the interface. This is due to the low solubility of the TMC in water and the fairly good solubility of MPD in organic solvent [5,6]. The thin PA top layer rejects various dissolved species, while the support layer provides the necessary mechanical strength for the membrane to withstand the high operating pressure [7,8]. Different polymers used for the preparation of the polymeric support layer are used. PSF polymer is an excellent material for the support layer, because it has excellent oxidative, thermal, and hydrolytic stability as well as good flexibility, resistance to extreme pH values, and good mechanical and film-forming properties [9,10]. The performance of a TFC membrane is primarily affected by two important factors: (1) The ultra-thin film chemistry, which is related to the property of the monomers utilized in the polymerization reaction; and (2) the reaction conditions used to synthesize the ultra-thin film on the porous support layer [11]. The performance of a PA-TFC membrane is affected by several parameters, including the preparation conditions of the thin upper layer via IP, feed concentration, temperature, flow rate, and feed pH. Many preparation conditions affect the structure and separation performance of PA-TFC membrane, such as reaction time and temperature, curing time and temperature, and monomer concentration [12,13,14]. Generally, optimization is used to improve the separation performance of PA-TFC membranes by obtaining higher water flux and rejection of salts without increasing the time and cost of production. One of the optimization methods during preparation is response surface methodology (RSM) [15]. RSM is a combination of mathematical and statistical techniques that can be used to develop, improve, and optimize processes. It can also be used to study the influence and interactions of several independent variables at different levels. The main objective of RSM is to optimize the output response based on the input variables investigated [16]. Many previous studies have used RSM to optimize the preparation conditions of PA-TFC membranes [17,18,19]. Currently, the novel advanced methods used for multi-objective optimization studies depend on a genetic algorithm, wherein RSM is used as a solver for this algorithm. Thus, the multi-objective optimization problem requires more than one objective to be optimized. In this case, one solution cannot satisfy both objective functions and the optimal solution of one objective may not be the best solution for other objectives. Therefore, the different solutions will produce trade-offs between different objectives, and a set of solutions is required to represent the optimal solutions of all objectives. Several methods and algorithms are used to determine a set of solutions from the given inputs during optimization, such as the multi-objective genetic algorithm II (MOGA-II). It is an efficient algorithm based on a new multi-search elitism method that combines random selection and directional crossover [20]. This new search method has the ability to preserve some excellent solutions and prevent premature convergence to local optima [21]. MOGA-II has become more popular due to its ability to approximate the set of optimal trade-offs in a single run [20]. Thus, MOGA-II has been used for achieving fast Pareto convergence [22].

In this work, we investigated various PA layer fabrication conditions to optimize PA-TFC membrane performance using a novel optimization method that combines the MOGA-II and factorial design methods. The input parameters used in this study were TMC concentration, reaction time, and curing temperature, and the output responses were water flux and rejection of salts with the maximization of both parameters being the main objective. Other responses, namely Na_2_SO_4_, MgCl_2_, and MgSO_4_ rejection, were determined as constraints during the optimization run. The experimental design used factorial design for the RSM. The MOGA-II optimization method utilized the response surfaces as the solver based on radial basis functions to predict responses for particular design points.

## 2. Materials and Methods

### 2.1. Materials

Polysulfone (PSF) polymer was kindly supplied by BASF (Berlin, Germany). N-methyl-pyrrolidone (NMP) was obtained from Oxford Lab Chem (Palghar, Maharashtra, India). Polyethylene glycol (PEG) 600was purchased from Merck (Darmstadt, Germany). Trimesoyl chloride (TMC, purity 98%) monomer was purchased from Merck (Kenilworth, New Jersey, USA), and m-phenylenediamine (MPD, purity 99%) monomers were purchased from Sigma-Aldrich. Trimethylamine (TEA, purity 99%) and hexane (C_6_H_14,_ purity 99%) were supplied by Oxford Lab Chem (Palghar, Maharashtra, India). Sodium dodecyl sulfate (SDS) was obtained from Sigma Aldrich (Kenilworth, New Jersey, USA). All salts (NaCl, MgCl_2_, Na_2_SO4_,_ and MgSO_4_) were purchased from Oxford Lab Chem (Palghar, Maharashtra, India). Deionized water (Milli-Q) with a resistivity of 18.2 MΩ cm was used in all of the experiments (Millipore, Temecula California, USA).

### 2.2. Fabrication of the Polysulfone (PSF) Support

PSF porous membrane was fabricated using the phase inversion method. First, PSF polymer was dried in an oven overnight at 50 °C. Casting solution was prepared by dissolving dried PSF (17 wt %) and PEG (15 wt %) in NMP solvent. The casting solution was cast on a glass plate using a casting knife film applicator (DeltaE Srl, Italy) to obtain a thickness of 90 μm. The glass plate was then immersed in a coagulation bath at room temperature to form the porous PSF support. To complete the phase inversion process, the formed PSF membrane was incubated in coagulation bath for 24 h. The membrane was then removed from the coagulation bath, cut, and prepared for TFC fabrication steps.

### 2.3. Preparation of a Thin Film Composite (TFC) Membrane

The TFC membranes were prepared by depositing a thin layer of polyamide onto the prepared PSF supports via interfacial polymerization (IP) between MPD and TMC monomers. The TFC membrane preparation process is shown in Figure 1. Likewise, the PSF support membrane was immersed in Milli-Q water overnight, removed from the water, and positioned on a plastic plate. A rubber gasket and a plastic frame were placed on top of the support membrane, and binder clips were used to hold the plate–membrane–gasket–frame stack together. The aqueous MPD solution (1 wt%) containing SDS (2 wt%) and TEA (1 wt%) was prepared in Milli-Q water. TEA was used as an acylation catalyst to promote the IP reaction by removing HCl, which is formed as a by-product during the reaction. SDS is used to increase absorption of the MPD solution into the PSF support [23,24]. MPD solution (30 mL) was poured onto the frame and allowed to contact the PSF support membrane for 10 min to ensure penetration of the MPD solution into the pores of the PSF support. The MPD solution was drained and the excess was removed from the surface using a rubber roller and air-drying for approximately 3 min until no droplets were observed. Different concentrations of TMC in n-hexane (0.1 and 0.2 wt%) were poured on the impregnated PSF support membranes to complete the IP reaction on the surface at different reaction times (15, 30, and 60 s). The TMC solution was drained from the frame, and the frame and gasket were disassembled. The synthesized TFC membranes were rinsed with hexane to remove unreacted monomers and cured in a digital oven at different temperatures (60 and 80 °C) for 5 min. Finally, the TFC membranes were prepared and stored in the refrigerator at 4 °C until use [1,25].

### 2.4. Characterization

The cross-section morphologies of the thin polyamide (PA) film composite TFC membranes were studied by field-emission scanning electron microscopy (FESEM; JEOL, Akishima, Tokyo Japan). The membrane surface topography was analyzed using an atomic force microscope (AFM; Veeco NanoScope V Multi Mode software). Differential scanning calorimetry (DSC) was used to study the crosslinking of the PA active layer. During the DSC analysis, the TFC samples were run under nitrogen atmosphere in a DSC 6000 (PerkinElmer). The flow rate of nitrogen gas was 20 mL/min, and the heating rate was 10 °C/min from 25 to 300 °C. To isolate the polyamide layer from the TFC structure, a dried sample of TFC membrane was incubated in dichloromethane to dissolve the PSF support layer. The PA active layer was washed with water and methanol, dried, and used for DSC.

### 2.5. Performance Evaluation of TFC Membrane

The performances of all of the membrane samples were examined using a pressure-driven filtration CF042 cross-flow cell filtration system (Sterlitech Corporation, Kent, WA, USA). The effective membrane surface area was 42 cm^2^. First, the membranes were compacted with Milli-Q water at a transmembrane pressure (TMP) of 8 bar for 10 h. Then, water flux was measured at different TMPs (ΔP) and calculated using Equation (1):(1)$$J_{w}=\frac{Q}{A∆t}$$
where, $J_{w}$ is the pure water flux (PWF) (Lm^−2^ h^−1^), Q is the volume of water permeated (L), A is the effective membrane area (m^2^), and ∆t is the permeation time (h).

Salt rejection of the TFC membranes was measured using a single-solute solution containing NaCl (2000 ppm), MgCl_2_ (1000 ppm), Na_2_SO_4_ (1000 ppm), and MgSO_4_ (1000 ppm). Filtration was performed at 8 bar with a flow rate of 1.3 L/min (LPM) at room temperature. The concentration of salt in the permeate was determined by measuring the electrical conductivity of the salt ions in the permeate using a conductivity meter (DeltaOhm HD 2156.1). The observed salt rejection (R) was calculated from the following Equation (2):(2)$$R\;\%=1-\;\frac{\sigma_{P}}{\sigma_{F}}\;\times 100$$
where $\sigma_{P}$ and $\sigma_{F}$ are the ion conductivity in the permeate and feed, respectively.

## 3. Optimization Study

### 3.1. Experimental Design

The experiments were conducted based on the factorial design method. Three parameters—TMC, Tc, and t—at different values were used for the optimization study as shown in Table 1. Analysis of variance (ANOVA) of the experimental results was performed using Minitab V17. The factorial design method and ANOVA analysis were used to determine the significant effects of the input parameters and their interactions on output responses. The ranges of the process parameters were selected based on previous studies as shown in Table 1.

### 3.2. Multi-Objective Optimization

The multi-objective genetic algorithm II (MOGA-II) optimization method was used to perform the optimization. In the MOGA-II method, several basic MOGA-II parameters are input by the user while the other parameters are internally set by the optimization program. Different functions, such as polynomial, neural network, and radial basis (Table 2), are used to generate the response surface model (RSM).

The function is chosen depending on the percent variation between the real and computed virtual designs points. Using Mode Frontier^®^, the evaluation study is conducted to determine which function should be used to generate the RSM, which is used in the MOGA-II method. The MOGA-II optimization process is as follows: (i) MOGA-II starts with an initial population set (P) that is used for the radial basis construction. (ii) The value of the objective function of the initial population is calculated. Thus, radial basis response surfaces are created for each output based on the first population and improved during simulation with the addition of new design points. (iii) MOGA-II generates a new population (P^’^) via selection, crossover, and mutation in which the radial basis is used as an evaluator to measure the error for each design point. (iv) If the error for a generated point is acceptable, it is included in the next population to be run through the MOGA-II algorithm. If the error is not acceptable, the point is used as a new design point for the radial basis construction. (v) MOGA-II converges when the maximum allowable Pareto percentage has been achieved, resulting in the optimized values as the output. (vi) However, if convergence does not occur, the process is repeated starting at step (iii) up to the maximum number of iterations [28]. The flow chart for the optimization process is shown in Figure 2.

Optimization was primarily conducted to maximize the water flux and NaCl rejection. The remaining three responses were considered as constraints in the optimization process. The objective functions and constraints are listed in Table 3. The workflow for the optimization using Mode Frontier^®^ is depicted in Figure 3.

## 4. Results and Discussion

### 4.1. Thin Film Composite Membrane Morphology

Figure 4 shows cross-sectional scanning electron microscope (SEM) images. The deposited PA active layer (100–300 nm thick) on the porous PSF substrate can be seen clearly in the SEM images. In addition, as shown in Figure 4a–c, by increasing the polymerization reaction time from 15 to 60 s, a thicker PA active layer formed due to the increased number of amine monomers diffusing to the organic phase [29]. The thicknesses of the PA active layers were 110, 212, and 259 nm for the 15, 30, and 60 s reaction times, respectively. In contrast, by increasing the TMC concentration from 0.1 to 0.2 wt%, the PA active layer thickness decreased to 199 nm due to the formation of a denser PA active layer, which hindered diffusion of MPD and prevented active layer growth [24,30,31] as shown in Figure 4d.

Figure 5 shows 3-D topographic AFM images of TFC membranes. The AFM results indicated that polymerization at the low reaction time (15 s) resulted in the production of a highly rough surface (Figure 5a). An increase in the polymerization time from 15 s to 60 s resulted in decreased surface roughness as shown in Figure 5b. This difference is owing to the fact that the reaction at low time occurred so fast that the interface between the two immiscible phases was diffusional. As the polymerization time increased, the PA active layer formed and became denser and thicker which inhibited diffusion of the MPD solution across the PA active layer to the TMC solution. This phenomenon is called ‘self-limiting’ and has been reported by Y. Jin et al. [32].Thus, the chemical structure may have changed the surface free energy, which changed the surface roughness and morphology [13]. In contrast, the roughness measured by AFM decreased with increasing TMC concentration from 0.1 to 0.2 wt% (Figure 5c) due to the rapid formation of a thinner and denser PA active layer which decreased the diffusion of MPD toward the TMC solution [33,34].

To estimate the effect of polymerization time on the degree of polyamide crosslinking, differential scanning calorimetric (DSC) was used. DSC of the PA active layer of TFC membranes and their glass transition temperature (Tg) are shown in Figure 6. Tg plays an important role in the degree of crosslinking and is directly proportional to the cross-linking density (φ); thus, ΔTg ≈ kϕ, where k is a material constant [35]. As shown in the results, increasing the polymerization time from 15 to 60 s increased the degree of crosslinking, and therefore, the glass transition temperature increased from 154 to 167 °C. This indicates that the PA active layer had more crosslinking. By increasing the polymerization time, the number of polyamide chains and networks of inter- and intramolecular hydrogen bonds increased, which resulted in a significant increase in the cohesive energy of the polymer that increased the Tg [36].

### 4.2. Analysis of Variance (ANOVA)

ANOVA was used to evaluate the significance of the interfacial polymerization reaction parameters (i.e., TMC, Tc, and t) on membrane performance parameters, including water flux and rejection of salts. The ANOVA results for the flux and NaCl rejection responses are shown in Table 4, which includes (i) the degrees of freedom (DF), (ii) the adjusted sum of squares (Adj SS), (iii) the adjusted mean squares (Adj MS), (iv) the F-value, and (v) the *p*-value. *P*-values < 0.05 indicate the parameters and interaction parameters that had a significant effect [37]. The significant parameters in the ANOVA table have been represented in italic text to simplify their readability. Based on the results in the ANOVA table, the two-way interactions of the reaction parameters did not significantly affect any of the responses. Furthermore, water flux was only significantly affected by TMC concentration and not influenced by the single parameters, Tc or t, nor their interaction (Table 4).

This is may be due to the effect of TMC concentration on the density or thickness of the PA active layer. An increase in the concentration of TMC lead to a decrease in the molar ratio of amine to acyl chloride (–NH_2_:–COCl), which resulted in a thinner (Figure 4d) and denser (Figure 5c) PA active layer with lower water flux [1,38]. The ANOVA results concerning NaCl rejection (%) indicate that TMC, Tc, and t have significant effects on NaCl rejection (%) compared to water flux, which was only impacted by TMC. Thus, these parameters (TMC, Tc, and t) are important for conferring the performance of the PA active layer with regard to NaCl rejection. The high TMC concentration has been found to induce formation of a denser PA film that can effectively reject salts [38].

Heat curing is often required to complete polymerization reaction and to help remove any residual organic solvent from the PA active layer [39]. During curing, the monomers diffuse into the interface for polymerization, which increases the extent of crosslinking on the thin PA film [40,41], and thereby salts rejection increased. In general, the reaction time (t) plays a significant role in determining the extent of polymerization—specifically, the cross-linking reaction extent between TMC and MPD, the thickness of the top PA active layer, and performance of the fabricated TFC membrane [40]. With the short reaction time, the PA active layer was thin (Figure 4a), and less crosslinking occurred (Figure 6; 15 sec); the prepared membrane thus exhibited low performance (i.e., decreased salt rejection). In contrast, the thickness (Figure 4c) and extent of crosslinking (Figure 6; 60 sec) of the PA active layer of the synthesized TFC membrane were higher with the increased reaction time, which resulted in increased salt rejection.

The effect ratios of the input parameters on the output responses is expressed in terms of relative strength, which were computed from the means of the smoothing spline ANOVA in Mode Frontier^®^. Figure 7 shows the relative strength of the significant input parameters and their interactions, and Figure 8 demonstrates the effects of the significant input parameters on all responses using the response surfaces for radial basis functions. As can be observed from Figure 7, the TMC concentration, t, and their interaction (TMC × t) are the most dominant factors affecting flux. Moreover, the relative strength value of TMC is higher than the other input parameters. The variation in flux as a function of TMC concentration and reaction time is shown in Figure 8a. At the low concentration and low reaction time, the water flux was higher. Increasing the TMC concentration led to decreased flux (Figure 8a; blue color), because the increased concentration contributed to the increased density of the PA active layer. In contrast, an active layer with low density was formed with high water flux when the low TMC concentration was used. Figure 7 shows that NaCl rejection is affected by all of the individual input parameters (i.e., TMC, t, and Tc) and their interactions except t*Tc. The effect of curing time was considerably higher than those of TMC concentration and reaction time.

As shown in Figure 8b, the NaCl rejection is directly proportional to the curing temperature at any concentration of TMC. Thus, the low NaCl rejection obtained at low curing temperature was because that temperature was insufficient for complete polymerization to occur. Figure 8c shows that NaCl rejection gradually increases with increasing curing temperature and reaction time. This is attributed to the fact that as the curing temperature increases, the extent of crosslinking increases, and this leads to the formation of a denser PA active layer, which has high NaCl rejection performance. Similarly, an increase in reaction time results in a thicker PA active layer, which also increases its NaCl rejection ability. The effect of reaction time is clear at the low curing temperature; with increasing reaction time, NaCl rejection increases. Thus, polymerization increases and results in a thicker PA active layer. Figure 7 shows the effects of all of the individual parameters and the TMC*Tc interaction on Na_2_SO_4_and MgSO_4_ rejection properties. The values are approximately the same for both Na_2_SO_4_ and MgSO_4_, likely because both salts are divalent and have similar molecular weights. The variation in Na_2_SO_4_ rejection values using different TMC concentration and curing temperature, and the variation in MgSO_4_ rejection values based on TMC concentration and reaction time are shown in Figure 8d,f, respectively. Generally, their rejection was high with the lowest rejection of Na_2_SO_4_ and MgSO_4_ being 96.6% and 95.8%, respectively. As is shown in Figure 8d,f, at high curing temperatures and high reaction times, the percent rejection to both salts are high and reach 99.8% and 99.3% for Na_2_SO_4_ and MgSO_4_, respectively. Figure 7 shows that the interaction, t*Tc, did not affect any of the output responses.

### 4.3. Optimization Study

Table 1 shows the experimental results of 12 runs obtained from factorial design for the selected input parameters and the corresponding output responses. First, evaluation of the functions to be used for the response surfaces model was conducted to determine the best function for the optimization study using Mode Frontier^®^. In the evaluation study, three functions, i.e., polynomial, neural network, and radial basis, were utilized. Figure 9 shows the RSM distance chart, which is a line chart showing the distance between the real and virtual design points computed with the RSM function. The real points represent the original experimental points while the virtual points correspond to the predicted points by RSM.

Virtual design points were closer to the real design points using the radial basis function as shown in Figure 9c. It was concluded that the radial basis function was the best function, which was subsequently used in the response surface method. Table 5 lists the R^2^ values for all of the evaluated functions and corresponding output responses.

The R^2^ value equaled 1 when using the radial basis function. This indicates that the radial basis function explains all the variability of the response data around its mean and is the function that best fits the data. Therefore, the MOGA-II optimization method utilized the response surfaces based on the radial basis function as the optimization solver to predict responses for the design points. The optimization problem was formulated to maximize water flux and NaCl rejection (%) of the PA-TFC membrane at the same time. The lower limits of the remaining performance responses were used as constraints. Values less than 98% for Na_2_SO_4_ rejection, 96% for MgCl_2_ rejection, and 98% for MgSO_4_ were considered unfeasible solutions during optimization. The salts rejection limitations were selected based on the characterizations of a nanofiltration membrane (NF) and reverse osmosis membrane (RO). Therefore, the divalent salt rejection value was greater than 98%. The optimization model (i.e., objective functions and the constraints) are listed in Table 3. The workflow for the optimization procedure was developed using Mode Frontier^®^ program as depicted in Figure 3. The total number of design points are equal to the design points in the DOE table multiplied by the number of generations. The characteristics of the total of design points obtained from the optimization runs using MOGA-II are displayed using bubble charts. Design points contain on original DOE matrix as real and predicted points from RSM as virtual. The unfeasible design points are those that violate the constraints in the optimization study as listed in Table 3. The bubble chart is a variant of the scatter chart that enables the simultaneous visualization of three or four dimensions where the total design points are plotted relative to two objective functions, i.e., flux and NaCl rejection, as shown in Figure 10a.

The TMC concentration is represented by the size of the bubbles, which indicates that higher NaCl rejection correspond with higher TMC concentrations. The design points with the lowest NaCl rejection values have moderate to high flux and low to moderate TMC concentrations. Figure 10a shows that the unfeasible design points are mostly characterized by moderate NaCl rejection, low to moderate water flux, and moderate to high TMC concentration. Since the objective of the optimization study was to maximize flux and NaCl rejection, the design points corresponding to the top right corner of the bubble chart were selected as the optimal solutions. A Pareto-front is shown in Figure 10a that connects the optimal design points from A to F. The 4D bubble chart shows four variables at a time, and is plotted against the two objective functions, i.e., flux and NaCl rejection (Figure 10b). Here, the diameter of the bubbles represents the curing temperature, whereas the color represents the reaction time. The effect of curing temperature on NaCl rejection is very obvious in the 4D chart, wherein high NaCl rejection area is associated with high curing temperature. These design points are characterized by the high reaction time and the different TMC concentrations in the user range. This results in denser crosslinking of the PA active layer at high curing temperatures, which led to a clear increase in NaCl rejection. Similar findings were reported by G. E. Chen et al. [40]. As shown in Figure 10b, a decrease of curing temperature of 80 °C to 60 °C had an obvious impact on NaCl rejection, while the water flux was not significantly affected.

Figure 10c shows a 4D bubble chart in which bubble size represents Na_2_SO_4_ rejection and color represents the MgSO_4_ rejection. The vast majority of the design points have high Na_2_SO_4_ rejection and high MgSO_4_ rejection.

Another way to show and analyze the relationships between all input and output variables is by using a parallel coordinate chart as shown in Figure 11. A parallel coordinate chart can show design points with all the parameters used in the study. The optimum design points are linked to high water flux and higher salt (i.e., NaCl, Na_2_SO_4_, MgSO_4_, and MgCl_2_) rejection. Optimum results were observed with high curing temperature, high reaction time, and different TMC concentrations within the range used in this study. The Tc and t parameters only significantly affect NaCl rejection, so the maximum NaCl rejection was found at high Tc and higher t. While TMC concentration affects flux and salts rejection, for this reason, different values of TMC concentration were appeared at optimum points. Two optimal design points correspond to the original DOE matrix and other optimal design points were generated with MOGA-II optimization technique. Some of the obtained optimum design points are shown in Figure 11 and are listed in Table 6. Of the seven optimal design points, design points A and H are real and correspond to the original DOE matrix. For high NaCl rejection, the design point, A, is considered the optimum point while design point H should be selected for high flux.

## 5. Conclusions

In the present work, a developed algorithm combing a radial basis response surface and the MOGA-II optimization algorithm was used to optimize the fabrication conditions of PA-TFC membranes. The experimental design based on the factorial design methodology was conducted to test the three fabrication parameters: TMC concentration, reaction time, and curing temperature. The ANOVA results table and response surfaces were used to determine the effects of fabrication parameters on membrane performance. The TMC concentration, reaction time, and their interaction (TMC × t) were the most dominant factors affecting water flux, the high flux was at 0.1 wt% TMC concentration, while NaCl rejection was affected by all of the input parameters (TMC, t, and Tc) and their interactions except (t*Tc), the high NaCl rejection was 97.71% at 0.2 TMC concentration, 60 s reaction time, and 80 °C curing temperature. The RSM evaluation results showed that the radial basis function was the best at predicting good results (R^2^ = 1) compared to the polynomial and neural network functions. Based on these results, the radial basis function was chosen to generate the RSM that was used as the solver for MOGA-II. The MOGA-II optimization algorithm was successfully used for the multi-objective optimization of the fabrication conditions for PA-TFC membranes by interfacial polymerization. The optimal solutions were characterized by high reaction time (60 s), high curing temperature (80 °C) and different values of TMC concentration to maximize water flux and NaCl rejection.

## Figures and Tables

**Figure 1 polymers-12-00494-f001:**
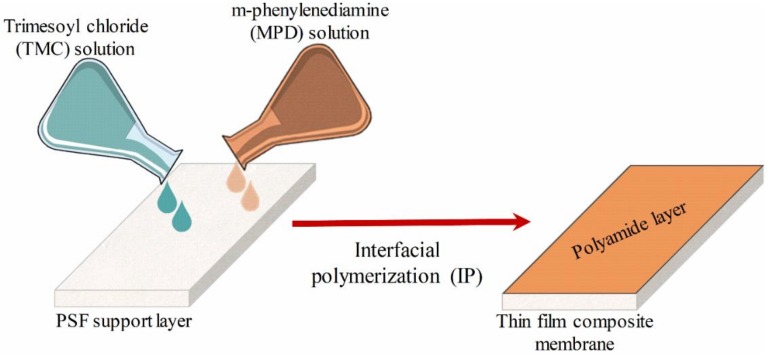
Protocol steps used to prepare polyamide-thin film composite (PA-TFC) membrane.

**Figure 2 polymers-12-00494-f002:**
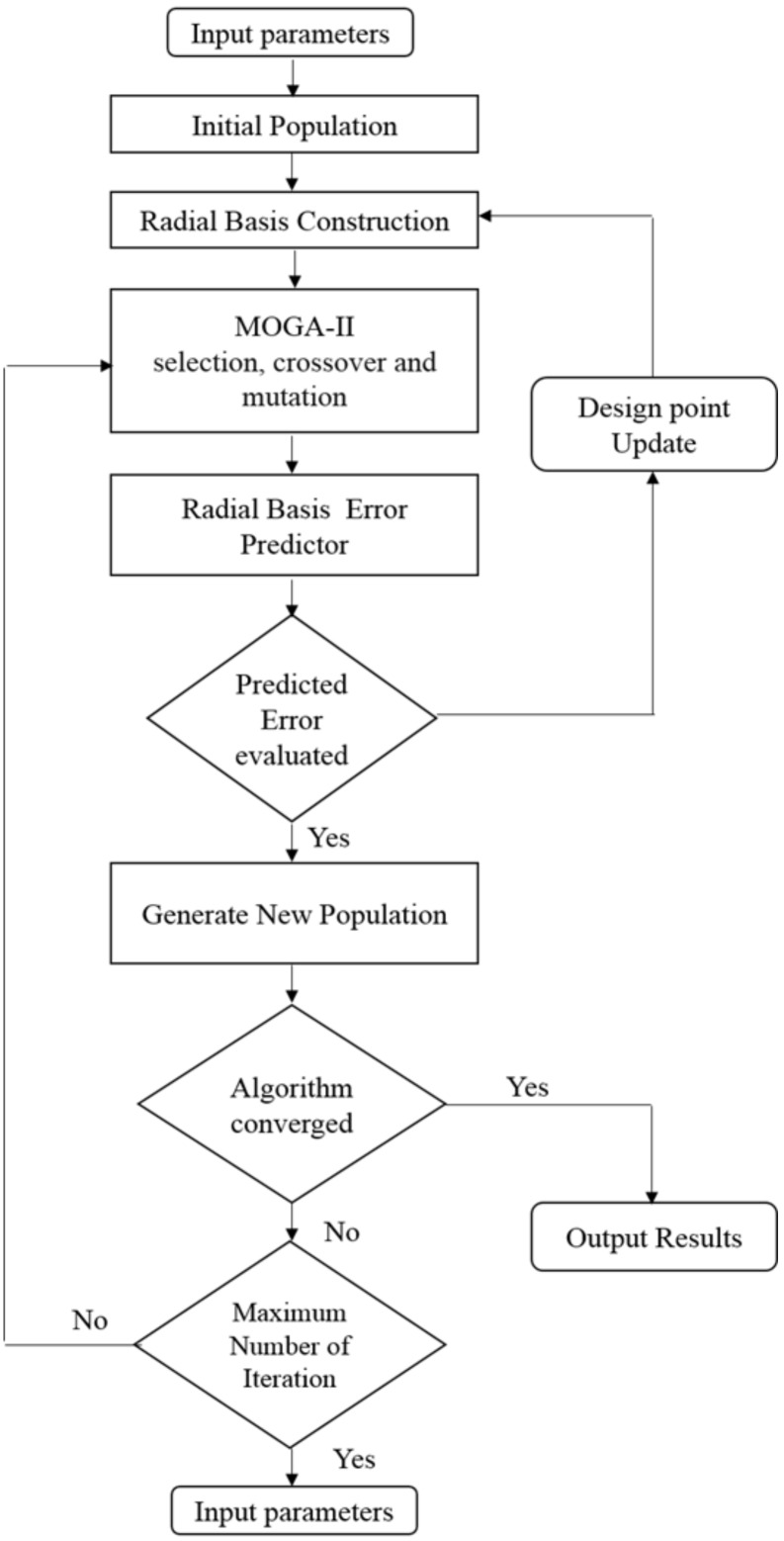
Flow chart of the multi-objective genetic algorithm (MOGA) optimization process.

**Figure 3 polymers-12-00494-f003:**
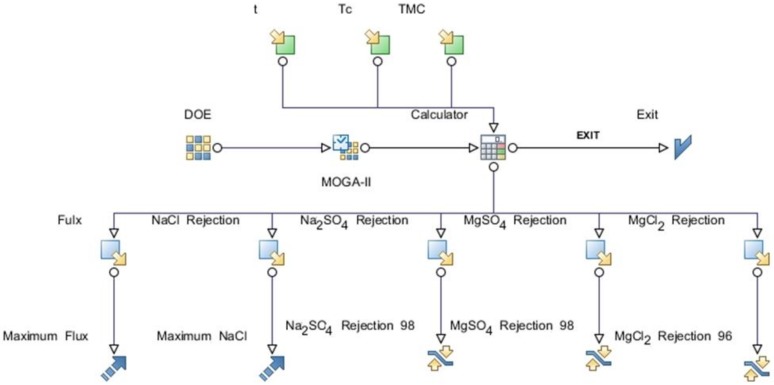
Optimization workflow using MOGA-II.

**Figure 4 polymers-12-00494-f004:**
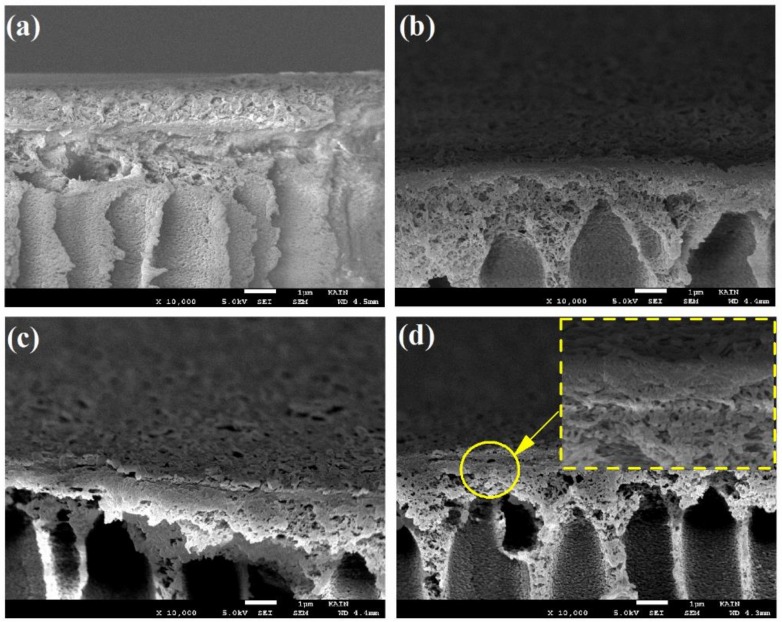
Scanning electron microscope (SEM) micrographs of the cross section for different PA-TFC membranes; (**a**–**c**) 0.1 trimesoyl chloride (TMC) and 15, 30, and 60 s, and (**d**) 0.2 TMC and 60 s.

**Figure 5 polymers-12-00494-f005:**
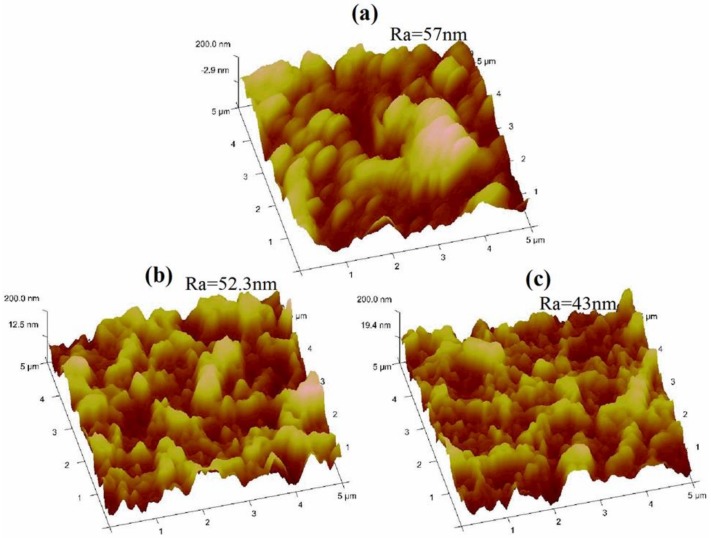
3D topographic atomic force microscope (AFM) images at different fabrication parameters; (**a**) 0.1 TMC and 15 s, (**b**) 0.1 TMC and 60 s, and (**c**) 0.2 TMC and 60 s.

**Figure 6 polymers-12-00494-f006:**
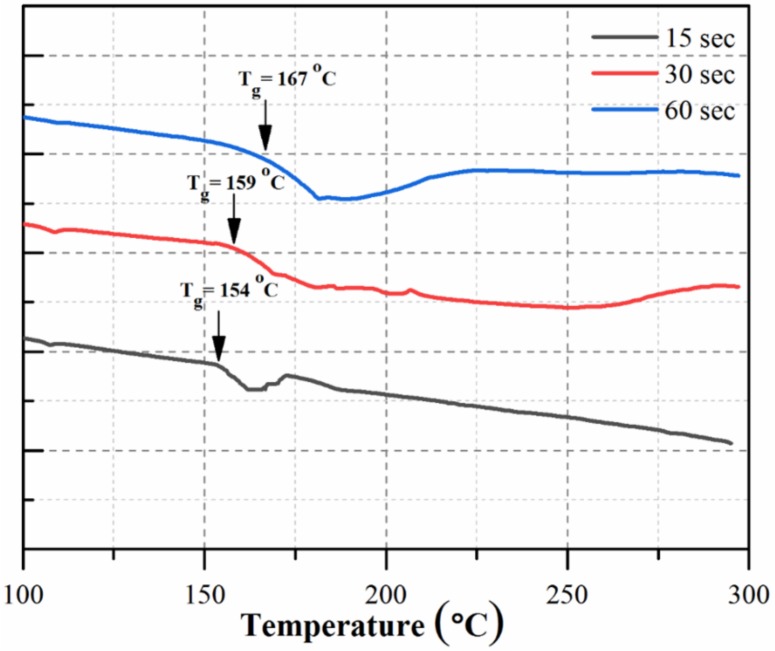
Differential scanning calorimetric (DSC) curves of polyamide (PA) layers at different reaction times (15, 30, and 60 s).

**Figure 7 polymers-12-00494-f007:**
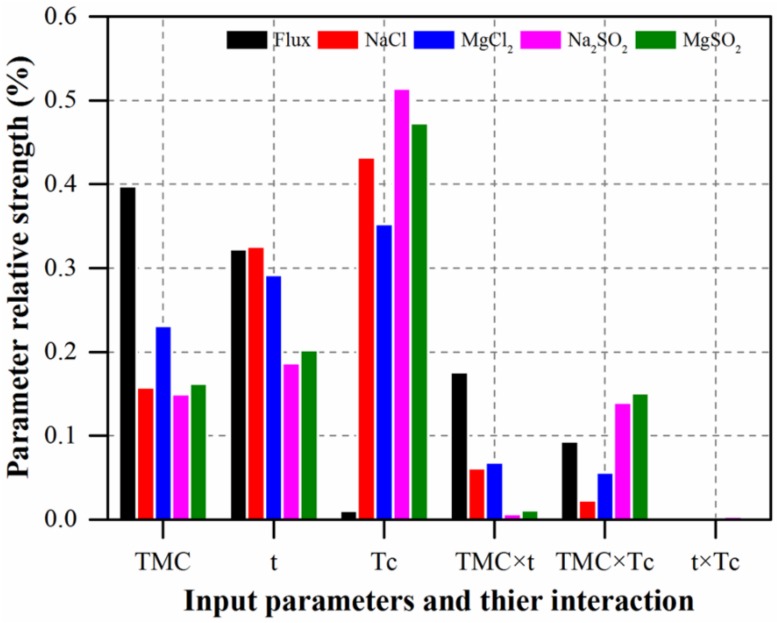
ANOVA spline relative strength of input variables and their interactions for all responses.

**Figure 8 polymers-12-00494-f008:**
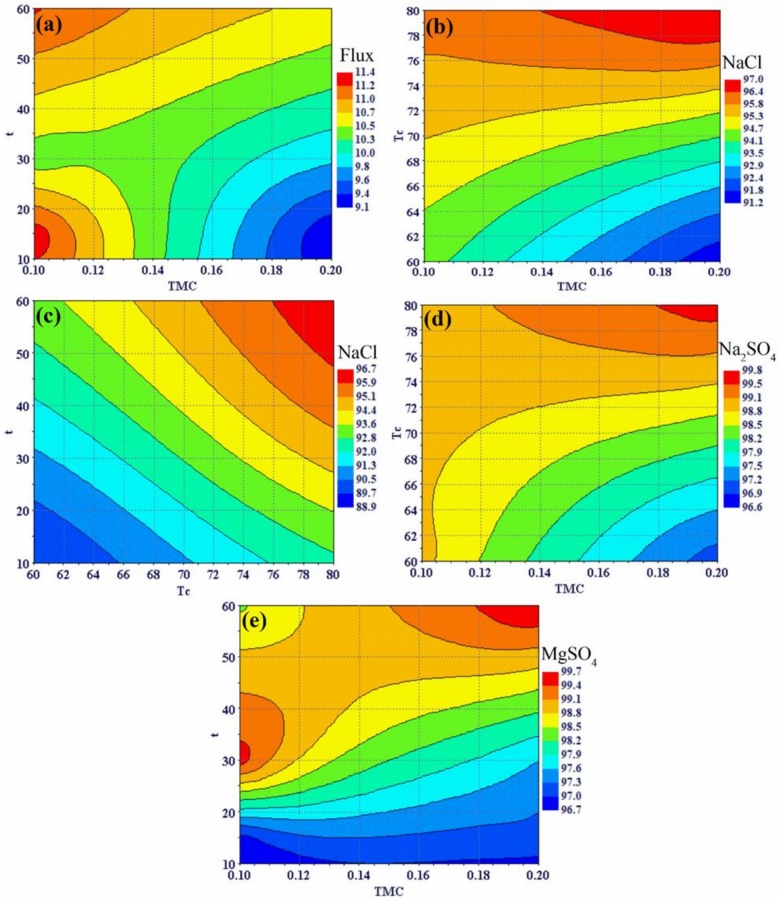
Response surface plots, effects of the fabrication parameters on (**a**) flux, (**b**,**c**) NaCl rejection, (**d**) Na_2_SO_4_ rejection, and (**e**) MgSO_4_ rejection.

**Figure 9 polymers-12-00494-f009:**
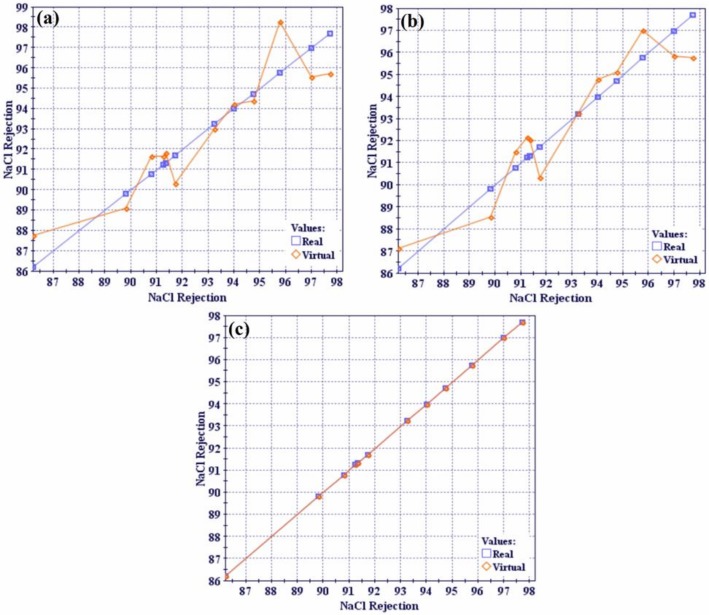
Response surface model (RSM) distance charts to used functions (**a**) polynomial, (**b**) neural network, and (**c**) radial basis.

**Figure 10 polymers-12-00494-f010:**
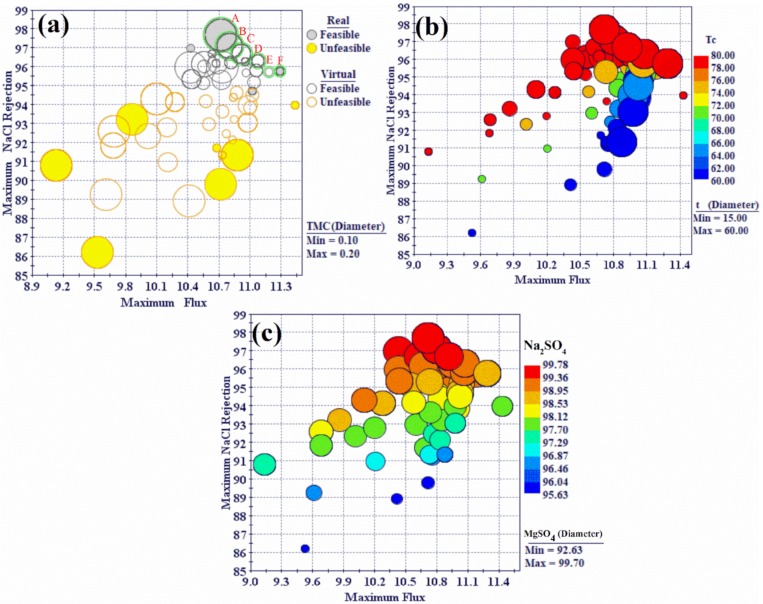
(**a**) 3D bubble chart showing the design points obtained with the two objectives (flux and NaCl rejection) and TMC concentration variable; (**b**) 4D bubble chart showing the design points obtained with two objectives of flux and NaCl rejection and two variables of Tc and t; and (**c**) 4D bubble chart showing the design points obtained with output variables of flux and NaCl Na_2_SO_4_, and MgSO_4_ rejections.

**Figure 11 polymers-12-00494-f011:**
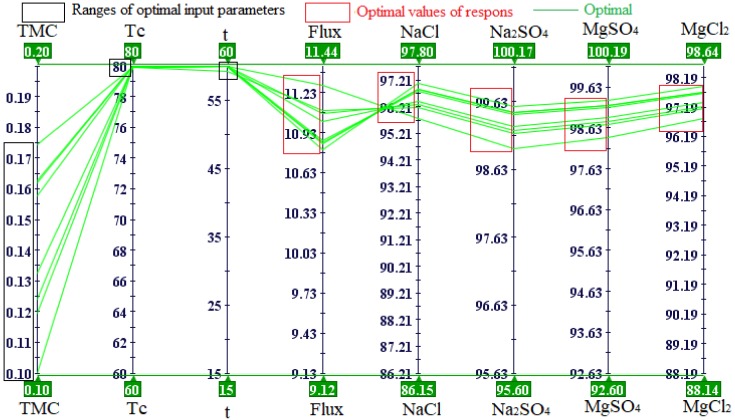
A parallel coordinate chart for the analysis of fabrication parameters for TFC membrane.

**Table 1 polymers-12-00494-t001:** Design of experiment (DOE) runs for the experimental plan.

Exp. No	Input Parameters			Responses				
	TMC (wt%)	t (sec)	Tc (^o^C)	Flux (l/m^2^.h)	NaCl Rejection (%)	MgCl_2_ Rejection (%)	Na_2_SO_4_ Rejection (%)	MgSO_4_ Rejection (%)
1	0.1	15	80	11.42	94	95.30	97.74	96.77
2	0.1	30	80	10.43	96.99	97.96	99.65	99.49
3	0.1	60	80	11.28	95.76	96.80	98.94	98.42
4	0.2	15	80	9.13	90.79	93.19	97.59	97.09
5	0.2	30	80	9.86	93.24	94.06	98.75	97.52
6	0.2	60	80	10.71	97.71	98.45	99.77	99.69
7	0.1	15	60	10.68	91.72	94.33	98.01	96.84
8	0.1	30	60	10.75	91.25	93.67	96.83	95.88
9	0.1	60	60	11.02	94.73	96.46	98.92	98.47
10	0.2	15	60	9.52	86.20	88.19	95.62	92.63
11	0.2	30	60	10.71	89.81	91.53	95.70	94.26
12	0.2	60	60	10.87	91.33	93.03	96.59	95.14

**Table 2 polymers-12-00494-t002:** Response surface functions.

Function Name	Function Form	Parameters	Ref.
Polynomial Regression	$y=\beta_{0}+\sum_{j=1}^{k}\beta_{j}x_{j}+\varepsilon$	β: Coefficient matrix x structural matrix k is the number of input factors ε is random error	[26]
Neural Network	$F\left(x\right)=\;\frac{1}{1+e^{-\mathrm{kx}}}$	x refers to input parameters	[26]
Radial Basis Function	$F\left(x\right)=\;\sum_{i=1}^{m}\beta_{i}\emptyset \|X-X_{i}\|$	X is input parameters $\beta_{i}$: Coefficient matrix $\emptyset \left(*\right)$: RBF function $\left\|X-X_{i}\right\|$: Euclidean norm	[27]

**Table 3 polymers-12-00494-t003:** Objective functions and constraints for the optimization study.

Objective Functions	Maximize Flux Maximize NaCl Rejection
Constraints	Na_2_SO_4_ > 98% MgCl_2_ > 96% MgSO_4_ > 98%

**Table 4 polymers-12-00494-t004:** Analysis of variance (ANOVA) table for flux and NaCl rejection.

Source	DF		Flux			NaCl Rejection			
		Adj SS	Adj MS	F-Value	*p*-Value	Adj SS	Adj MS	F-Value	*p*-Value
Model	6	4.293	0.7156	3.52	0.094	108.186	18.0310	9.23	0.014
Linear	3	2.900	0.967	4.76	0.063	95.470	31.8233	16.29	0.005
TMC	1	1.579	1.579	7.78	*0.039*	16.448	16.4479	8.42	0.034
t	1	1.2804	1.280	6.30	0.054	33.960	33.9602	17.39	0.009
Tc	1	0.041	0.041	0.20	0.673	45.062	45.0619	23.07	0.005
Two-Way Interaction	3	1.067	0.356	1.75	0.272	8.789	2.9296	1.50	0.322
TMC × t	1	0.698	0.698	3.44	0.123	6.416	6.4163	3.29	0.130
TMC × Tc	1	0.368	0.368	1.81	0.236	2.371	2.3714	1.21	0.321
t × Tc	1	0.000	0.0005	0.00	0.962	0.001	0.0010	0.00	0.983
Error	5	1.015	0.203			9.766	1.9532		
Total	1	15.309				117.952			

**Table 5 polymers-12-00494-t005:** Comparison of response surface functions for NaCl rejection.

No.	Functions	Mean Absolute Error	Mean Relative Error	Mean Normalized Error	R-Squared
1	Neural network	9.54 10^−1^	1.03 10^−2^	8.30 10^−2^	0.885
2	Polynomial SVD	1.01	1.09 10^−2^	8.80 10^−2^	0.843
3	Radial basis	0	0	0	1

**Table 6 polymers-12-00494-t006:** Optimum design points.

ID	TMC	t	Tc	flux	NaCl	MgCl_2_	MgSO_4_	Na_2_SO_4_
A	0.2	60	80	10.71	97.712	98.45	99.77	99.69
B	0.18	60	80	10.77	97.21	97.99	99.61	99.4
C	0.16	60	80	10.85	96.82	97.65	99.45	99.16
D	0.15	55	80	10.915	96.69	97.556	99.392	99.07
E	0.126	56.7	80	11.003	96.28	97.22	99.22	98.82
F	0.1	60	80	11.28	95.8	96.8	98.93	98.42

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
