# Peer review of "A Novel Approach to Optimize the Fabrication Conditions of Thin Film Composite RO Membranes Using Multi-Objective Genetic Algorithm II"

_polymers, 2020, doi:10.3390/polym12020494_

Round 1
Reviewer 1 Report
The manuscript "A novel approach to optimize the fabrication conditions of thin film composite RO membranes using Multi-objective Genetic Algorithm II " revealed some characterizations and modeling of of polyamide (PA) thin film composite (TFC) membranes using the Multi-objective Genetic 15 Algorithm II (MOGA-II) method. So far the results interesting but which lacks a little are how its presented.
The main issue of this script is the missing goal why such modeling important and which application would it suit for. please include that.
The other point refer to certain presentation as example Figure 1 showing how its made doesn,t give much info and should shown in sheme more than individual handlings
Page 2 line 24: What is a Pareto convergence? please explain.
The script is written for experts in modeling and sometimes difficult to follow
As example Figure 9 what is real and what virtual and why in Figure 9a and b those virtual jump so differently. The reviewer suggest to give more explanation how such values effect the composite membranes having a direct link in RO applications.
Figure 10-12 can be concluded in one figure here as well should be given more explanations how the material used benefits such results.
Figure 13 is very difficult to understand due to the presentation of multicurves, please give more explanation in text and find some other presentation form.
Somehow there no mean values with standard deviation shown, are only one sample made or several?
The conclusion gives mainly the results of modeling but no results of real testings of their membranes shown in characterization and usuage. The conclusion don,t reflect the manuscript and need be modified.
Author Response
Query (1): The main issue of this script is the missing goal why such modeling important and which application would it suit for. please include that.
Response: Thank you very much for the valuable suggestion. MOGA-II is an efficient algorithm based on a new multi-search elitism method that combines random selection and directional crossover. This new search method has the ability to preserve some excellent solutions and prevent premature convergence to local optima. MOGA-II has become more popular due to its ability to approximate the set of optimal trade-offs in a single run. Hence, MOGA-II is used for achieving fast Pareto convergence.
In addition, the main applications of MOGA-II which would it suit for RO membrane development are; 1) reduce cost of material by reduction of number of experiment for membrane fabrication; 2) saving time; 3) generation of prediction results in high amounts; 4) and can able to study many objectives together in respect to many responses in single run. Hence, MOGA-II is optimization method, and is tested on several multi-objective optimization problems.
Query (2): The other point refer to certain presentation as example Figure 1 showing how its made doesn,t give much info and should shown in sheme more than individual handlings
Response: Thank you for this comment, as per reviewer valuable suggestion, Figure 1 is changed.
Query (3): Page 2 line 24: What is a Pareto convergence? please explain.
Response: Multi-objective genetic algorithm (MOGA) is an optimization technique. In MOGA a set of solutions called Pareto front is obtained. Due to the random nature of MOGA, finding an optimal solution is a difficult and time-consuming task. Pareto convergence is a decision method to find the optimal solution among all possible solutions in Pareto front. In revised manuscript reference related to Pareto convergence is added.
Query (4): Figure 9 what is real and what virtual and why in Figure 9a and b those virtual jump so differently. The reviewer suggests to give more explanation how such values effect the composite membranes having a direct link in RO applications.
Response: Authors are agreed with this scientific comment. It is explained in that; the real points represent the original experimental points while the virtual points correspond to the predicted points by RSM. This explanation has been added in the manuscript in page 12 line 342-343.
Regarding the second part of the reviewer's comment. In this study, three models have been tested namely “polynomial, neural network, and radial basis functions for development the response surface for the outputs. I was found that, due to the errors mentioned in Table 5, the predicted points (virtual) obtained from the neural network and the polynomial models did not match with the actual (real) points. That is why in figure 9, the virtual points differ from the real points. The values of R2 in Table 5 also confirm that the two models (neural network and polynomial) were not suitable for developing the response surfaces.
Generally, the values obtained from simulation and modeling predict the effect the composite membranes quality, such as thickness, surface roughness, and polyamide crosslinking, all of these parameters have a direct link in RO applications.
Query (5): Figure 10-12 can be concluded in one figure here as well should be given more explanations how the material used benefits such results.
Response: Authors agree with the reviewer to combine Figures 10 – 12 into one Figure (Figure 10).
Such results are very significant for example, one solution cannot satisfy both objective functions and the optimal solution of one objective may not be the best solution for other objectives. Therefore, the different solutions will produce trade-offs between different objectives, and a set of solutions is required to represent the optimal solutions of all objectives. Several methods and algorithms are used to determine a set of solutions from the given inputs during optimization, such as the MOGA-II. It is an efficient algorithm based on a new multi-search elitism method that combines random selection and directional crossover. This new search method has the ability to preserve some excellent solutions and prevent premature convergence to local optima. Also, another benefit of this, can be seen clearly in figure 10; where the bubble chart is a variant of the scatter chart that enables the simultaneous visualization of 3 or 4 dimensions where the total design points are plotted relative to two objective functions, i.e., flux and rejection, with different responses in single figure.
Query (6): Figure 13 is very difficult to understand due to the presentation of multicurves, please give more explanation in text and find some other presentation form.
Response: As per reviewer suggestion, manuscript is revised by putting the modified figure 10 (before figure 13), also, explanation is added in text.
Query (7): Somehow there no mean values with standard deviation shown, are only one sample made or several?
Response: Thank you for this comment, actually, all experiments are repeated three times, and average value is reported in manuscript.
Query (8): The conclusion gives mainly the results of modeling but no results of real testings of their membranes shown in characterization and usuage. The conclusion don,t reflect the manuscript and need be modified.
Response: As per reviewer suggestion, in revised manuscript, the conclusion is modified, reflecting the manuscript.
Reviewer 2 Report
The authors developed a novel algorithm to optimize the fabrication conditions of polyamide (PA) thin film composite (TFC) membranes. Three different fabrication conditions were chosen, TMC concentration, reaction time and curing temperature, for investigating the optimal solutions and dominant factors affecting performance of PA-TFC membranes. However, there are still some deficiencies in the format and content of the paper as the follows:
- There are only two~ three sets of data for each factor, TMC concentration (0.1 and 0.2 wt. %), reaction time (15, 30, and 60 s) and curing temperature (60 and 80°C), is that enough to establish an accurate mathematical model? And there are many factors affecting the interfacial polymerization, it is difficult to control, whether the theory has effective significance to guide the experiment.
- Some literature can enrich the content of the introduction part like Journal of Membrane Science, 452 (2014) 82; Journal of colloid and interface science 560 (2020) 273.
- The innovation of this paper is to propose a new algorithm to optimize the reaction conditions, what is the main improvement compared to the traditional algorithm, and whether more accurate results can be obtained.
- It is difficult to see the change of PA layer density from Fig.4d., please clarify the thickness change here.
- In this paper, “by increasing the polymerization reaction time from 15 to 60 s, a thicker PA active layer formed due to the increased number of amine monomers diffusing to the organic phase” and “This difference is owing to the fact that the reaction at low time occurred so fast that the interface between the two immiscible phases was diffusional.” Please explain the specific effect of reaction time on two-phase diffusion.
- The SEM image and AFM images, some of the changes are not well explained, the analysis is not detailed and accurate enough, and the description process is not rigorous enough. Please refer to Chemical Communications 56 (2020) 478; Journal of Membrane Science 597 (2020) 117753; ACS Applied Materials & Interfaces 7 (2015) 9534for the correction and further analysis.
- “Figure 5. 3D topographic AFM images at different fabrication parameters (a) 0.1 TMC and 15 sec, (b) 243 0.1 TMC and 60 sec, and (d) 0.2 TMC and 60 sec.”Where is the Fig.5(d)?
- There are some typo in this MS.
Author Response
Query (1): There are only two~ three sets of data for each factor, TMC concentration (0.1 and 0.2 wt. %), reaction time (15, 30, and 60 s) and curing temperature (60 and 80°C), is that enough to establish an accurate mathematical model? And there are many factors affecting the interfacial polymerization, it is difficult to control, whether the theory has effective significance to guide the experiment.
Response: Thank you for this valuable comment. Generally, in MOGA, a number of points are generated within the levels of the selected factors. For example, the lower level of the temperature factor (T) is 60 and the upper limit is 80, within these two values (60 and 80) the MOGA software can generate infinite numbers. So, the mathematical model was built based on a number of experiments plus the data generated by MOGA.
Query (2): Some literature can enrich the content of the introduction part like Journal of Membrane Science, 452 (2014) 82; Journal of colloid and interface science 560 (2020) 273.
Response : In revised manuscript, some literature is added to enrich the content of the introduction part.
Query (3): The innovation of this paper is to propose a new algorithm to optimize the reaction conditions, what is the main improvement compared to the traditional algorithm, and whether more accurate results can be obtained.
Response: Thank you for this comment on MOGA II. In our study, MOGA II is practiced since it has a good ability for making a global search and finding the optimal solution fast. It also does not need more input data. Moreover, as per the Author's knowledge, no study has been found in the optimization of reaction conditions by using the MOGA method for optimizing the fabrication conditions of PA-TFC membranes.
In the traditional algorithm, one solution cannot satisfy both objective functions and the optimal solution of one objective may not be the best solution for other objectives. Therefore, the different solutions will produce trade-offs between different objectives, and a set of solutions is required to represent the optimal solutions of all objectives. MOGA-II is an efficient algorithm based on a new multi-search elitism method that combines random selection and directional crossover. This new search method has the ability to preserve some excellent solutions and prevent premature convergence to local optima. MOGA-II has become more popular due to its ability to approximate the set of optimal trade-offs in a single run.
Query (4): It is difficult to see the change of PA layer density from Fig.4d., please clarify the thickness change here.
Response: Thank you for a suggestion, in revised manuscript, we added the high resolution SEM image with clarify the thickness.
Query (5): In this paper, “by increasing the polymerization reaction time from 15 to 60 s, a thicker PA active layer formed due to the increased number of amine monomers diffusing to the organic phase” and “This difference is owing to the fact that the reaction at low time occurred so fast that the interface between the two immiscible phases was diffusional.” Please explain the specific effect of reaction time on two-phase diffusion.
Response: This is excellent comment, and for the thin film composite membrane development, reaction time play very significant role. Phase diffusion is initially high which results in fast polymerization reaction, there by PA polymer chain forms fast. With increase time, diffusion of number of amine monomers decrease slightly because the formed PA polymer chain create resistance to diffusion, however PA polymerization reaction is continued with time, that is why thickness of PA layer increases in slow rate compare with the initial diffusion time. Hence, the reaction at low time occurred so fast and higher time number of amine monomers diffusion increases, which result in the formation of a thicker PA active layer.
Query (6): The SEM image and AFM images, some of the changes are not well explained, the analysis is not detailed and accurate enough, and the description process is not rigorous enough. Please refer to Chemical Communications 56 (2020) 478; Journal of Membrane Science 597 (2020) 117753; ACS Applied Materials & Interfaces 7 (2015) 9534for the correction and further analysis.
Response: As per reviewer suggestion, in revised manuscript, we explanted the SEM image and AFM images analysis followed through refer articles and cited in the specific text.
Query (7): “Figure 5. 3D topographic AFM images at different fabrication parameters (a) 0.1 TMC and 15 sec, (b) 243 0.1 TMC and 60 sec, and (d) 0.2 TMC and 60 sec.”Where is the Fig.5(d)?
Response: In revised manuscript, we update the Figure 5.
Query (8): There are some typo in this MS.
Response: Thank you for taking care our manuscript deeply. MS is read by all authors to avoid typo errors.